# Bisphenol A Adsorption on Silica Particles Modified with Beta-Cyclodextrins

**DOI:** 10.3390/nano12010039

**Published:** 2021-12-23

**Authors:** Stefan Bucur, Aurel Diacon, Ionel Mangalagiu, Alexandra Mocanu, Florica Rizea, Adrian Dinescu, Adi Ghebaur, Aurelian Cristian Boscornea, Georgeta Voicu, Edina Rusen

**Affiliations:** 1Faculty of Chemistry, Alexandru Ioan Cuza University of Iasi, 11 Carol 1st Bvd, 700506 Iasi, Romania; bucurm.stefan@gmail.com (S.B.); ionelm@uaic.ro (I.M.); 2Faculty of Applied Chemistry and Materials Science, University Politehnica of Bucharest, 1- 7 Gh. Polizu Street, 011061 Bucharest, Romania; aurel_diacon@yahoo.com (A.D.); mocanu_alexandra85@yahoo.com (A.M.); flori_rizea@yahoo.com (F.R.); ghebauradi@yahoo.com (A.G.); cristian.boscornea@yahoo.com (A.C.B.); getav2001@yahoo.co.uk (G.V.); 3Institute of Interdisciplinary Research—CERNESIM Centre, Alexandru Ioan Cuza University of Iasi, 11 Carol I, 700506 Iasi, Romania; 4National Institute for Research and Development in Microtechnologies (IMT-Bucharest), 126 A, Erou Iancu Nicolae Street, 011061 Bucharest, Romania; adrian.dinescu@imt.ro; 5Advanced Polymer Materials Group, University Politehnica of Bucharest, Gh. Polizu Street, 011061 Bucharest, Romania

**Keywords:** beta-cyclodextrin, bisphenol A, adsorption, covalently modified silica

## Abstract

This study presents the synthesis of silica particles bearing two beta-cyclodextrin (BCD) (beta-cyclodextrin-BCD-OH and diamino butane monosubstituted beta-cyclodextrin-BCD-NH_2_). The successful synthesis of the BCD-modified silica was confirmed by FT-IR and TGA. Using contact angle measurements, BET analysis and SEM characterization, a possible formation mechanism for the generation of silica particles bearing BCD derivatives on their surface was highlighted. The obtained modified silica displayed the capacity to remove bisphenol A (BPA) from wastewater due to the presence of the BCD moieties on the surface of the silica. The kinetic analysis showed that the adsorption reached equilibrium after 180 min for both materials with qe values of 107 mg BPA/g for **SiO_2_-BCD-OH** and 112 mg BPA/g for **SiO_2_-BCD-NH_2_**. The process followed Ho’s pseudo-second-order adsorption model sustaining the presence of adsorption sites with different activities. The fitting of the Freundlich isotherm model on the experimental results was also evaluated, confirming the BCD influence on the materials’ adsorption properties.

## 1. Introduction

Cyclodextrins (CDs) are a class of three-dimensional (3D) cyclic oligosaccharides composed of 6, 7, or 8 D-glucopyranosyl units linked by α-D-(1→4) bonds with amphiphilic characteristics given by the hydrophilic surface and the hydrophobic internal hollow [1]. These properties, correlated with non-toxic and biodegradable properties and the possibility to obtain CDs at the industrial level, led to a wide variety of applications tremendously useful for supramolecular chemistry, catalysis, chromatography, medicine, cosmetics, pharmacy, food, the perfume industry, and environment decontamination [1,2,3,4,5,6,7].

In terms of air, soil, or water decontamination, CDs’ properties were exploited by creating host–guest inclusion complexes based on the hydrophobicity of the inner cavity that facilitates the encapsulation of targeted molecules [8]. Different technologies were employed to remove various pollutants so far, but, according to Gong et. al. [9], adsorption processes are preferred compared with membrane technology, advanced oxidation processes, photocatalytic degradation, or extraction due to its higher efficiency and lower costs [10,11]. Regardless of the decontamination procedure, two strategies have been remarked in terms of CDs’ derivatives manufacturing, one based on its linking to polymer chains, while the other involved the CDs’ grafting onto carbonaceous or inorganic compounds [6,12,13,14,15,16,17,18].

Many CDs-based polymers have been used to remove organic pollutants from contaminated waters for higher absorption efficiency, but recycling procedures were proven to be limited sometimes since the structural integrity of the polymer chains gave rise to problems after repeated recovery cycles [19]. On the other hand, CDs-based inorganic nanoparticles’ (CDs-iNPS) complexes have attracted the attention of scientists since the synthesis of iNPS permits a high control over the size distribution, surface area, and the possibility to keep the intrinsic magnetic, photocatalytic, optical, or antimicrobial properties of the Inps [16,20]. These advantages of the hybrid CDs-iNPS complexes makes them a versatile platform for materials science, catalysis, sensors, nanomedicine, and decontamination processes [21,22,23,24].

Ever since bisphenol A (BPA) was discovered as a synthetic estrogen in 1890, its application as monomer in the plastic industry was intensively developed in food packaging, epoxy and polyacrylate resins, unsaturated polyesters, polysulfones, or polycarbonate plastics [25]. Although it is one of the most abundant additives used in plastics’ manufacturing, in the last decades strong attention has been given to the health effects towards exposure for adults and infants [26]. Thus, the exposure to BPA revealed that it can affect the functions of thyroid, the pancreas, the central nervous systems, the reproductive system, and the immune response of the human body [27,28]. BPA releases into the environment are related to the food chain and improper recyclability of plastic materials. Thus, the presence of BPA intake by dermal exposure or inhalation is inevitable [29]. According to the European Chemicals Agency and REACH regulations (Annex XVII-art. 66), BPA is classified as a chemical compound that negatively affects the fertility and hormonal systems of humans and animals. It damages eyes, may cause allergic skin reactions and respiratory irritation [30], and, since 2020, it has been replaced with Bisphenol S in thermal paper [31]. Even if BPA is replaced with Bisphenol S or Bisphenol F in certain applications, the problem of contaminated environment with BPA or its possible substitutes remains an international issue [32].

Different methods have been applied for the removal of BPA from contaminated environments involved in membrane separation processes, advanced oxidation procedures, or biological treatments [33,34]. The drawbacks of these methods are strongly related sometimes with the rigorous control of parameters in the case of biological or advanced oxidation procedures and laborious or expensive synthesis methods for membranes [35,36]. These circumstances have driven the attention of researchers to the design of CDs-based inclusion complexes that could be used to adsorb the BPA molecules from the polluted environment [37,38,39,40].

Silica-based materials have been used in numerous applications including pharmacy, medicine, catalysis, food, paints, and coatings [41,42,43,44] and recently in separation technologies with CDs [16]. The interest for silica-based materials is related to the ease of manufacturing and the characteristics of the final materials that can exhibit high specific surface areas (as high as 1500 m^2^/g) or porous structures depending on the synthesis method [45]. Another great advantage of using silica (SiO_2_) comes from the possibility to modify its surface with functional groups that enhance the adsorption process of targeted molecules inside the CDs’ cavities. For instance, byproducts from the incomplete fuel combustion, namely, polycyclic aromatic hydrocarbons (PAHs) were removed from water using two types of CDs (hydroxypropyl-β-cyclodextrin (HPBCD) and β-cyclodextrin (BCD)) and tetraethyl orthosilicate (TEOS) as SiO_2_ precursor [46]. The silica NPS functionalized with HPBCD and BCD facilitated the adsorption of PAHs’ hydrophobic molecules up to 1.65 mg/gm which was two folds higher compared with non-functionalized SiO_2_ nanoparticles [46]. Another study, developed by Carvalho et. al. [47], involved the synthesis of a hybrid CDs-SiO_2_ composite by the surface functionalization of SiO_2_ with BCD that was used to remove methyltestosterone hormone from contaminated waters. Methacryloyl-BCD was used also to functionalize SiO_2_ NPS to remove p-nitrophenol from contaminated waters, with kinetic studies confirming that fast adsorption processes occur in the case of BCD-SiO_2_-based composites and that the adsorption mechanism is determined by the concentration of the organic pollutants and the structure of the hybrid CDs-based material [48].

Thus, the novelty of our work involves the use of BCD-modified SiO_2_ particles for enhanced BPA removal from contaminated water. The BCD-SiO_2_ derivatives are aimed to improve the interaction between the pollutant and the adsorbent. Therefore, our goal was to provide a facile synthesis route to access SiO_2_-BCD materials and to investigate the maximum complexation capacity of the BCD-SiO_2_ hybrid composites towards BPA. Thus, adsorption kinetics, complexation mechanism, and isotherms were taken into consideration for our study.

## 2. Materials and Methods

Pyridine (Aldrich) was dried on molecular sieves of 3Å. The β-cyclodextrin (BCD-OH) (≥95.0%, Wacker Chemie) was vacuum dried before use for 24 h. Glucose (Aldrich anhydrous), phenol (Merck), sulfuric acid (Sigma-Aldrich, St. Louis, MO, USA), bisphenol A (Merck), ethanol (Chimopar), 3-(triethoxysilyl)propyl isocyanate, tetraethyl orthosilicate (TEOS) (Aldrich), cetyltrimethylammonium bromide (CTAB), and sodium hydroxide (Fluka) were used as received.

### 2.1. Synthesis of Diamino Butane Monosubstituted BCD (**BCD-NH_2_**) 

The synthesis procedure and characterization of **BCD-NH_2_** were presented in our previous study [39]. Briefly, the procedure for obtaining **BCD-NH_2_** was as follows: 5.5 g mono tosylated-BCD (4.266 mmol) were dissolved in 166 mL of 1,4-diaminobutane (DAB), slowly warmed up to 70 °C, and kept at this temperature for 24 h. At the end of the reaction time, the solvent was vacuum distilled and the solid that resulted was dissolved in a minimum amount of water. This syrup was added dropwise into 150–200 mL of acetone, which precipitated a white solid. At least two acetone precipitations are needed to obtain a white powder solid; otherwise, the product seems oily. The final product was obtained by drying the sample in a vacuum oven at 40 °C for 2 days with a 54% yield. NMR and MALDI characterization were presented in our previous study [39].

### 2.2. Synthesis of **Si-BCD-OH** and **Si-BCD-NH_2_**


One mmol of BCD derivative (1.14 g BCD-OH or 1.2 g BCD-NH_2_) was dissolved in 20 mL of dry pyridine by stirring at 70 °C. After 2.5 h of stirring, 1.1 mmol (0.27 mL) 3-(triethoxysilyl) propyl isocyanate were added to the BCD solutions. After 24 h, the solvent was removed by vacuum evaporation and the residue was washed three times with n-hexane. The obtained yield was over 95% for both intermediaries. The product was used directly without any other purification for the synthesis of SiO_2_-BCD-OH or SiO_2_-BCD-NH_2_ (please see Figure 1).

### 2.3. Synthesis of **SiO_2_-BCD-OH** and **SiO_2_-BCD-NH_2_**

One-half g of cetyltrimethylammonium bromide (CTAB) was dissolved in 160 mL of water. The solution was heated at 90 °C and 2 mL of NaOH 2 mol/L were added and stirred for 15 min. Then, 60 mmol (13.4 mL) TEOS were added while continuously stirring the reaction mixture. After 10 min from the introduction of TEOS (the solution became opalescent), the silane-BCD precursor (Si-BCD-OH or Si-BCD-NH_2_) (1 mmol) was introduced, and the reaction was stirred for an additional 4 h at 90 °C. After, the SiO_2_-BCD-OH and SiO_2_-BCD-NH_2_ were isolated by centrifugation and washed thoroughly with diluted HCl solution (0.01 M), water, ethanol, and acetone. The final products were dried at 70 °C for 24 h, affording around 4.2 g for each material (yield 85%).

Additionally, the SiO_2_ sample that was used during the characterization was obtained by the same procedure without the addition of a BCD silane derivative.

### 2.4. Methods

The morphologies of the materials were investigated through field emission gun scanning electron microscope (FEGSEM) Nova NanoSEM 630 (FEI) (Hillsboro, OR, USA).

A KSV CAM 200 apparatus was used for static contact angle measurements performed on dried films. Water droplets or CH_2_I_2_ were used with a drop volume of 20 µL. The measurement of each contact angle was made within 10 s after each drop to ensure that the droplet did not soak into the film. The contact angles reported were the mean of 10 determinations. To prepare the SiO_2_ and BCD-modified SiO_2_ films, solutions (5% weight) in isopropanol were prepared by sonication and then deposited by drop-casting on microscope glass plates (five successive deposition/drying cycles) to ensure a complete coverage of the glass plates.

FT-IR spectra were recorded on a Bruker VERTEX 70 (Billerica, MA, USA) spectrometer using 32 scans with a resolution of 4 cm^−1^ in 4000–600 cm^−1^ region. The samples were analyzed using the attenuated total reflection (ATR) technique.

The values for the specific surface areas of the modified SiO_2_ particles were determined with a gas (N_2_) porosimeter-type Gemini V based on the BET method.

The UV-Vis spectra were recorded using a V-550 Able Jasco spectrophotometer, using a bandwidth of 1 nm and a scanning speed of 1000 nm min^−1^. The BCD content was quantified by determining the reducing sugars of the silica using concentrated H_2_SO_4_ acidolysis and phenol colorimetric analysis [49]. The method requires the preparation of a calibration curve. Thus, solutions containing various concentrations of anhydrous glucose were mixed with 1 mL of 4% phenol standard solution and 7 mL of concentrated sulfuric acid. After vigorous shaking, the solutions were incubated at 50 °C for 30 min and then the absorbance of the aqueous mixture was measured at room temperature. The absorbance values at 490 nm were plotted versus the concentration of anhydrous glucose to obtain the calibration curve. The analysis procedure involved the dispersing of a 10-mg sample (**SiO_2_-BCD-OH** or **SiO_2_-BCD-NH_2_**) in 2 mL H_2_O. Then, 7 mL of concentrated sulfuric acid and 1 mL of 4% phenol standard solution were added and the solution was incubated at 80 °C to ensure hydrolysis and phenol coupling reaction. The glucose level was assessed using the calibration curve, and the BCD content was calculated as follows (Equation (1)):
(1)$$BCD\;content=\;\frac{c\times V\times M}{180\times n\times 0.01}\times 100\;\%$$
where c is the glucose concentration (g/L), V is the volume of mixed solution (L), M is the molar mass of BCD (g/mol), and n is the number of glucoses in a BCD unit.

The fluorescence spectra were registered using a FP-6500 Able Jasco spectrofluorometer. The adsorption of bisphenol A onto BCD-modified SiO_2_ (**SiO_2_-BCD-OH** and **SiO_2_-BCD-NH_2_**) was conducted using batch equilibrium technology in a mixed solution of ethanol and doubly distilled water (*v:v* 3:7) at the desired concentrations and a pH value of 5. In general, a 0.01-g sample of modified silica was dispersed thoroughly in 50 mL of solution of bisphenol A at various concentrations (5–30 mg/L) and shaken in a thermostatic bath (shaker operated at a speed of 180 rpm at 25 °C). After equilibrium was reached, the solid sample was separated by centrifugation. The concentration of bisphenol A in the residual solution was determined using fluorescence spectrophotometry (bisphenol A: λex/λem 274/307). The amount adsorbed onto the BCD-modified polymer spheres was determined using Equation (2):
(2)$$q_{e}=\frac{\left(c_{0}-c_{e}\right)\times V}{m}$$
where V is the solution volume (mL), c_0_ (mg L ^−1^) and c_e_ (mg L ^−1^) are the initial and final solution concentrations of bisphenol A, and m is the mass of BCD-modified polymer particles (mg).

The thermogravimetric analyses (TGA) were performed using Netzsch TG 209 F3 Tarsus equipment considering the next parameters: nitrogen atmosphere flow rate, 20 mL min^−1^; samples mass, ~3 mg; temperature range, room temperature −700 °C; and heating rate, 10 °C min^−1^ in an alumina crucible.

## 3. Results

The first aim of this study was the synthesis of silica particles modified with **BCD-OH,** respectively, **BCD-NH_2_**. The synthesis approach involved the synthesis of the BCD silane derivative by reacting the **BCD-OH** and **BCD-NH_2_** with 3-(triethoxysilyl) propyl isocyanate (TESPIC) (step A) followed by the hydrolysis–condensation together with TEOS (step B) (Figure 1 and detailed in Section 2.1 and Section 2.2, respectively).

To confirm the successful synthesis of the BCD silane derivative, FT-IR analysis was performed on the obtained compounds and materials (Figure 1A).

In the case of **Si-BCD-OH,** the characteristic bands of the hydroxyl and carbonyl groups can be noticed in the 3373 and 1726 cm^−1^ regions. The band observed at 2938 cm^−1^ was attributed to symmetric stretching of the C-H bond with carbonyl. Additionally, the specific bands for the urethane functional group 1736, 1608, 1570, and 1520 cm^−1^ were present. These bands are typical of the stretching C=O and N-H bonds. In the case of **Si-BCD-NH_2,_** the intensification of the signal at 1608 cm^−1^ specific for N-H vibration was detectable. After the BCD silane derivatives’ synthesis, the generation of silica particles involved the hydrolysis/condensation in the presence of TEOS. The FT-IR analysis was also employed to confirm the presence of BCD derivatives chemically attached to the silica (Figure 1B). Thus, we noticed the presence of the bands at 3400 cm^−1^ assigned to asymmetrical stretching of –OH groups, 2920 cm^−1^ associated with the vibration of the C-H stretch, the bands at 1154 cm^−1^ and 1030 cm^−1,^ owing to the vibrations of the asymmetric stretch of the C–O–C and symmetric stretching link C–O–C, respectively [50]. Further, silica-based materials displayed the characteristic signals for Si-O-Si vibration at 1080 cm^−1^. Thus, the FT-IR data confirmed the synthesis of silica modified with BCD derivatives covalently attached.

For further information about the degree of modification of the silica with BCD derivatives, thermogravimetric analysis was performed (Figure 2). Comparing the weight loss profiles of the two synthesized materials, it was noticed that they followed the same profile and there was only a slight difference between the samples. Thus, the degree of modification with BCD was similar, with a slightly higher value being registered for the **SiO_2_-BCD-OH**. The weight loss up to 120 °C was attributed to the physisorbed water desorption from the surface and pores [51], while at temperatures of over 200 °C the weight loss was due to de-hydroxylation of silica surface [52]. At temperatures of 280–320 °C, the decomposition of BCD took place [39]. It was noticed that the weight loss for **SiO_2_-BCD-OH** was around 29.7%, while for **SiO_2_-BCD-NH_2_** the value was 25.8%. However, these values included the remaining templating agent CTAB.

To more accurately determine the amount of BCD attached to the silica, UV-Vis analysis was employed. Thus, the determined BCD content was 23.97% (weight %) and 23.49 (weight %) for **SiO_2_-BCD-OH** and **SiO_2_-BCD-NH_2_**, respectively. Thus, the degree of modification of the silica with BCD was almost identical, which can be explained by the high reactivity of the -NCO towards -NH_2_ and -OH.

The morphological investigation of the materials was performed by scanning electron microscopy (Figure 3).

From the SEM images, the spherical shape of the SiO_2_ particles was visible, but their dimension varied between the samples. In the case of **SiO_2_-BCD-OH**, the particle diameter was about 300–350 nm, while for **SiO_2_-BCD-NH_2_** and **SiO_2_** it was around 150–200 nm. The silica generation process included several steps involving the formation of the initial “clouds” followed by densification, resulting in the primary particles that underwent collapse or association, leading to assemblies that undertook the growth process [53]. In our case, the particle size variation was attributed to the alteration of the particle growth process due to the presence of Si-BCD-OH derivatives (Figure 2). This is plausible since the initial nucleation stage was independent of the presence of Si-BCD derivatives; therefore, the initial particles’ number and primary particles should be similar in all cases. However, the insertion of Si-BCD changed the growth process, and, in the case of **Si-BCD-OH**, the growth of the assemblies was favored. In contrast, for **Si-BCD-NH_2_**, the process was much more similar to **SiO_2_**. In the case of **Si-BCD-OH**, there was the chance of di-silyl derivatives being formed as the result of similar reactivity between the -OH groups. In contrast, in the case of **Si-BCD-NH_2_**, due to the higher reactivity of -NH_2,_ the monosubstituted silyl formation was favored. Thus, the presence of di-silyl **Si-BCD-OH** may be the explanation for the alteration of the silica formation process by promoting the growth of the assemblies.

For further information about the characteristics of the modified SiO_2_ particles, BET analysis was performed. The BET analysis afforded a surface area of 81, 53, and 90 m^2^/g for SiO_2_ unmodified, **SiO_2_-BCD-OH,** and **SiO_2_-BCD-NH_2_**, respectively. Thus, the analysis of the results indicated that the surface area in the case of **SiO_2_-BCD-NH_2_** was almost double compared to **SiO_2_-BCD-OH** and only slightly higher than the unmodified SiO_2_. The results sustain the mechanism of particle generation and are in accordance with the SEM analysis.

To ascertain this assumption, contact angle measurements towards water were performed on the unmodified silica and on the two BCD-modified samples. The obtained contact angles values were 19° (**SiO_2_**), 16° (**SiO_2_-BCD-NH_2_**), and 14°(**SiO_2_-BCD-OH**). Due to the small difference between the samples, we also performed contact angle measurements using CH_2_I_2_. The values for CH_2_I_2_ were 11° (**SiO_2_**), 15° (**SiO_2_-BCD-NH_2_**), and 27° (**SiO_2_-BCD-OH**). Thus, it was noticed that the hydrophilic characteristic of SiO_2_ increased [54] with the addition of the BCD derivatives covalently attached to the silica due to the increased number of hydroxyl groups from the glucose units.

Exposure to BPA is a concern because of the possible health effects on the brain and prostate gland of fetuses, infants, and children [26]. Therefore, it is of an outmost importance to develop an efficient method of BPA trapping from aqueous solutions. In this study, the capacity of SiO_2_ particles functionalized with BCD derivatives to remove BPA from aqueous solutions was explored. Thus, the next step of our study consisted of the determination of the adsorption capacity at equilibrium and, respectively, the time required to reach equilibrium state. The adsorption profiles of the two BCD functionalized silica are presented in Figure 4.

After 180 min of interaction at 25 °C using a bisphenol A solution of 100 mg/L (Figure 4), an equilibrium state was reached for both materials with a q_e_ value of 107 mg BPA/ g for **SiO_2_-BCD-OH** and 112 mg BPA/ g for **SiO_2_-BCD-NH_2_**. From Figure 4. it can be noted that the absorption process can be divided into two steps. First is a rapid adsorption stage, which can be explained by the high bisphenol A concentration in solution that diffuses and is adsorbed to the polymer particles’ surface. The second stage, after approximatively 150 min, represents an intermediary behavior when the adsorption rate decreases as the internal diffusion resistance increases, leading to the equilibrium state. To evaluate the adsorption process, several mathematical models were employed to determine its efficiency and the mechanism that it follows: Lagergren’s pseudo-first-order kinetic model (Equation (3)), Ho’s pseudo-second-order model (Equation (4)), and Weber’s intraparticle diffusion model (Equation (5)):
(3)$$\ln\left(q_{e}-q_{t}\right)=lnq_{e}-k_{1}t$$
(4)$$\frac{t}{q_{t}}=\frac{1}{k_{2}q_{e}^{2}}+\frac{t}{q_{e}}$$
(5)$$q_{t}=K_{p}\sqrt{t}$$
where q_e_ (mg g ^−1^) and q_t_ (mg g ^−1^) are the amounts of bisphenol A adsorbed per unit mass of BCD-modified SiO_2_ (**SiO_2_-BCD-OH** and **SiO_2_-BCD-NH_2_**) at equilibrium and t (minutes), respectively, k_1_ is the pseudo-first-order adsorption rate constant (min ^−1^), k_2_ is the pseudo-second-order adsorption rate constant (g (mg min)^−1^, and K_p_ is the intraparticle diffusion constant (mg g^−1^ min^−0.5^).

From the analysis of Figure 5A,B and Table 1 data, it was noticed that, although the high value for R^2^, the Lagergren pseudo-first-order adsorption model failed to explain the materials’ response during the adsorption process, since the q_e, calc_ value was very different than the experimental. The Lagergren pseudo-first-order model assumes that the rate of change of solute uptake with time is directly proportional to the difference in saturation concentration and the amount of solid uptake with time, which is generally applicable over the initial stage of an adsorption process. Therefore, we opted to investigate the suitability of Ho’s pseudo-second-order adsorption model (Figure 5C,D). The pseudo-second-order kinetic model assumes that the rate-limiting step is chemical sorption or chemisorption and predicts the behavior over the whole range of adsorption. Thus, the adsorption rate is dependent on adsorption capacity not on concentration of adsorbate. The linear fitting for Ho’s model (Equation (2)) afforded good fitting, with a R^2^ of 0.921 and 0.959 for **SiO_2_-BCD-OH** and **SiO_2_-BCD-NH_2_**, respectively. Additionally, the determination of q_e, calc_ afforded a value of 127.22 mg/g and 131.23 mg/g for **SiO_2_-BCD-OH** and **SiO_2_-BCD-NH_2_**, respectively, which are only slightly higher than the experimental results. Therefore, Ho’s pseudo-second-order adsorption model could be used to explain the materials’ adsorption capacity. This would sustain the critical influence of the BCD moieties on the materials’ properties during the adsorption process, confirming its participation in the process through the introduction of the BPA in the BCD inner cavity. The fitting of Weber’s intraparticle diffusion model was also explored with results comparable to those of Ho’s pseudo-second-order adsorption model (Figure 5 E,F). However, considering that a good dispersion of the materials was achieved in water, this model is not the best suited to explain the adsorption mechanism for the material as a dispersed phase in water.

The study of the adsorption isotherms offers information on the interaction between the adsorbate and the adsorbent when the adsorption process reaches equilibrium and allows the determination of the adsorption capacity of the material, which is an important parameter for system evaluation. The most intensively used isotherm adsorption models are Langmuir (Equation (6)) and Freundlich (Equation (7)).
(6)$$\frac{1}{q_{e}}=\frac{1}{q_{max}}+\frac{1}{q_{max}K_{L}}\times \frac{1}{c_{e}}$$
(7)$$lnq_{e}=lnK_{F}+\frac{1}{n}lnc_{e}$$
where *q_e_* (mg g^−1^) indicates the amount of adsorbate at equilibrium, *q_max_* (mg g^−1^) indicates the maximum amount of adsorbate at equilibrium, *c_e_* (mg L^−1^) is the equilibrium concentration of the adsorbate in the solution, *K_L_* (L mg^−1^) and *K_F_* (L mg^−1^) are the Langmuir and Freundlich constants, respectively, and *n* is the heterogeneity factor.

The linearization of the two selected isotherms in Equations (4) and (5) and the parameters calculated are presented in Figure 6 and Table 2. From Figure 6A,C, it can be noticed that, although there is good fit for the linearization (R^2^ 0.949 and 0.994) of the Langmuir isotherms (Equation (6)), the value for the intercept is negative, resulting in negative values for K_L_ and q_max_ for **SiO_2_-BCD-OH** and **SiO_2_-BCD-NH_2,_** confirming that the Langmuir isotherm is not well suited to explain the adsorption at equilibrium for both materials. Since the Langmuir isotherm does not apply to the materials’ adsorption at equilibrium, it means that the monolayer adsorption of BPA did not take place. The Langmuir model can used to describe most of the adsorption processes that involve predominantly chemical interactions between solute and sorbent and, with some restrictions, for physical sorption processes. The primary condition for this model validity is the existence of only one type of active site on the surface of the sorbent [55]. Thus, the results suggest that the adsorption sites on the materials exhibited a heterogeneity in terms of activity, which can be explained by a higher activity of the BCD during the adsorption process compared to the Si-OH binding sites present on the silica. Furthermore, the preferred adsorption of BPA on the BCD moieties was sustained by the fitting of the Freundlich isotherm, which is usually specific for chemisorption adsorption processes. Unlike the Langmuir model, the Freundlich isotherm’s general hypothesis are (1) not all the active centers on the surface of the sorbent are energetically equal; (2) there may be a certain interaction between the solute molecules, and, therefore, once the surface of the sorbent is covered, additional molecules can still be adsorbed; and (3) the model can be used to describe the adsorption process of solute in multilayer [56]. The results obtained from the linearization of the Freundlich isotherm for 1/n values >1 are indicative of S-type isotherms. These are relatively uncommon but are often observed at low concentration ranges for compounds containing a polar functional group, at low concentrations. Such compounds are in competition with water for adsorption sites [57]. In this case, there was a competition between water and BPA adsorption on Si-OH sites and at the interior of the BCD cavities. *K_F_* is the Freundlich constant, which represents adsorption capacity as well as the *n* term, which is an empirical constant indicating the adsorption intensity of the system was larger for **SiO_2_-BCD-NH_2,_** indicating both a stronger adsorption and higher capacity of the material, probably due to the presence of amino functionality, which led to a different access to the BCD inner cavity.

Comparing the data presented in Table 3, it was observed that the obtained materials in this study displayed good adsorption capacity towards bisphenol A. This can be explained by the steric morphology of the particles, which facilitated high adsorption due to a high surface/volume ratio due to the particle size and BCD content. Although there was only a small difference between the maximum amount of BPA that can be adsorbed on the materials, the interaction strength between BPA and the adsorbent was distinct. Thus, the **SiO_2_-BCD-NH_2_** displayed a stronger interaction, as suggested by the slightly shorter period until the equilibrium state was attained and from *K_F_* and *n* values of the Freundlich isotherm.

## 4. Conclusions

This study presents the synthesis of silica modified with two BCD derivatives (BCD-OH and BCD-NH_2_). The obtained modified silica displayed the capacity to adsorb organic molecules such as bisphenol A due to the presence of the BCD moieties on the surface of the silica. The chemical attachment of the BCD derivatives (BCD-OH and BCD-NH_2_) to the surface of the silica particles was qualitatively confirmed by FT-IR spectroscopy and quantitatively by acidolysis and phenol colorimetric analysis. The degrees of silica functionalization using the BCD-OH and BCD-NH_2_ derivatives were 23.97 and 23.49 weight %. The morphology of the obtained silica particles was investigated by SEM and BET analyses. Using contact angle measurements, a possible formation mechanism for the generation of silica particles bearing hydrophile groups on their surface was highlighted.

The kinetic analysis of the adsorption revealed that the process reached equilibrium after 180 min for both materials, with q_e_ values of 107 mg BPA/ g for **SiO_2_-BCD-OH** and 112 mg BPA/ g for **SiO_2_-BCD-NH_2_.** The kinetics followed Ho’s pseudo-second order adsorption model, sustaining the critical influence of the BCD moieties on the materials’ properties during the adsorption process and its participation in the process through the introduction of the BPA in the BCD inner cavity. This aspect was also demonstrated by the fitting of the Freundlich isotherm model on the experimental results. The materials’ adsorption capacity and the process kinetics make them suitable candidates for wastewater treatment processes.

## Data Availability

Not applicable.

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
