# Peer review of "Bisphenol A Adsorption on Silica Particles Modified with Beta-Cyclodextrins"

_nanomaterials, 2021, doi:10.3390/nano12010039_

Round 1

Reviewer 1 Report

This work prepared beta-cyclodextrin modified SiO2 particles and used for removal of bisphenol A.  This manuscript is well written and organized.

1.  Please briefly introduce the synthesis procedure and characterization of BCD-NH2 in Section 2.1.

2.  It is hard to measure contact angle for particle sample. Please introduce the procedure to prepare the film sample from particles.

Reviewer 2 Report

The manuscript by Rusen et al. presents an interesting approach to attach cyclodextrin moieties onto silica nanospheres (prepared by the Stoeber method) which are then used for the adsorption of bisphenol-A from aqueous media.

The menuscript is well written, but some points have to be clarified:  

page 8, and Fig. 1B: the conclusion, that the attachement of cyclodextrine to SiO2 NP can be proven from FTIR spectra are not convicing. The FTIR spectra of neat SiO2 and modifies SiO2 are very similar, and the FTIR signals around 2900 cm-1 are found in both samples. 

page 9: here ist reads "the samples were digested". This is unclear. Obviously the text refers to the analytical procedure based on H2SO4 / phenol to analyze the glucose content.  

page 11: a detailed procedure for conatct angle measurements on the powdery materials is missing. The reported differences in water contact angle are low and do not provide sigificant information on the success of the modification of SiO2 nanoparticles.                                                                 

# the chapter on adsorption kinetics of BPA on cyclodextrin functionalized NP  is too long, and can be shortened.

# pages 17 + 18 and Table 3: numerous materials based on cyclodextrin have been reported in the literature with respect to adsorption of bisphenol A as pollutant. However, the capacity of the material presented by the authors has comparatively low adsorption capacity. So the question remains: what is the adavantage of the nes SiO2-cyclodextrin nanoparticles compared to other materials ? Are their advantages with respect to industrial application, wastewater purification and so on ? This shouls be discussed in the section "Conclusion".

# Reference 39 is identical with Reference 49.

Reviewer 3 Report

Interesting experimental work proposing solution to environmental chemistry. Nice graphics.

I propose acceptation  with minor revision (Improve abstract and conclusions).

Reviewer 4 Report

The manuscript describes the synthesis of hybrid nanomaterials based on silica surface modified with cyclodextrin ligands. Although the subject is of great importance, the manuscript in its present state cannot be published in nanomaterials, on the one hand because of the editorial part to be revised and on the other hand because it lacks data to apprehend the interest of these materials.

The manuscript is difficult to read because there is a lot of identical information everywhere in the text and in particular the bibliography on the toxicity of BPA as well as the different materials already used. An introduction with almost 50 references is not a good sign in terms of conciseness and precision. The whole text needs to be revised in terms of writing. Same remark for some figures such as figure 1 where the spectra are not very visible because one under the other and no attribution on the spectrum performed.

Then concerning the scientific data, some important data are cruelly missing such as the yields of syntheses and the obtained masses, the specific surfaces of these nanomaterials by BET and the obtained micro or nanostructure.

For those presented, more detailed explanations must be made. For example, the SEM images show a heterogeneity of shapes, to what is this attributable? The TGA curves still decrease at 700°C, why did authors stop at this temperature?

How to explain the formation of linear structure in scheme 2 in basic medium when it is known that these conditions favor 3D structures?

Are the differences observed in the contact angle measurements significant?

Why modeling the same phenomenon throughout the entire adsorption  experiment when the authors mention two adsorption processes?

Are the results in ATG and UV concordant? can the authors give a crude formula of these compounds? How to validate the monosubstitution of the compound Si-BCD-OH?

On page 16 it is mentioned "BCD inner cavity" what does it mean and how to quantify it?

minor errors :

BPA is not explained in the summary

No reference to figure 1 in chapter 2.3

Round 2

Reviewer 4 Report

Despite my remarks about a thorough rewriting of the manuscript to make it more readable, little evolution between the two versions has been done. There is still as much repetition and a cruel lack of data on the microstructure of these nanomaterials. How to rigorously evaluate the efficiency of a nano-absorbent without knowing its specific surface? How to deduce from the TGAs a percentage of relative loss of the organic when a plateau has not been reached? No elemental analysis either? no possible discussions on the two adsorption mechanisms because of the global study?
As far as I am concerned, this manuscript is still not up to the standard of a publication in nanomaterials.
